# Fermented *Sargassum fusiforme* Mitigates Ulcerative Colitis in Mice by Regulating the Intestinal Barrier, Oxidative Stress, and the NF-κB Pathway

**DOI:** 10.3390/foods12101928

**Published:** 2023-05-09

**Authors:** Siteng Zhang, Yu Cao, Zixuan Wang, Huanhuan Liu, Yue Teng, Guopeng Li, Jiaxiu Liu, Xiaodong Xia

**Affiliations:** 1State Key Laboratory of Marine Food Processing and Safety Control, National Engineering Research Center of Seafood, School of Food Science and Technology, Dalian Polytechnic University, Dalian 116034, China; 20171083200817@xy.dlpu.edu.cn (S.Z.);; 2College of Food Science and Nutritional Engineering, China Agricultural University, Beijing 100083, China

**Keywords:** *Sargassum fusiforme*, inflammatory bowel disease, proinflammatory cytokines, intestinal barrier

## Abstract

In recent years, *Sargassum fusiforme* has gained increasing attention for its ability to improve human health and reduce the risk of disease. Nevertheless, there have been few reports on the beneficial functions of fermented *Sargassum* fusiforme. In this study, the role of fermented *Sargassum fusiforme* in the mitigation of ulcerative colitis was investigated. Both fermented and unfermented *Sargassum fusiforme* demonstrated significant improvement in weight loss, diarrhea, bloody stools, and colon shortening in mice with acute colitis. Fermented *Sargassum fusiforme* further protected against goblet cell loss, decreased intestinal epithelium permeability, and enhanced the expression of tight junction proteins. Fermented *Sargassum fusiforme* reduced oxidative stress, which was demonstrated by a decrease in nitric oxide (NO), myeloperoxidase (MPO), and malondialdehyde (MDA) concentrations in the colon of mice and an increase in total superoxide dismutase (T-SOD) activity in the colon. Meanwhile, catalase (CAT) concentrations in both the colon and serum of mice were significantly increased. Fermented *Sargassum fusiforme* also attenuated the inflammatory response, which was evidenced by the decreased level of pro-inflammatory cytokines in the colon. Moreover, fermented *Sargassum fusiforme* inhibited the nuclear factor-κB (NF-κB) signaling pathway and increased the production of short-chain fatty acids in the intestine. These findings indicate that fermented *Sargassum fusiforme* may have the potential to be developed as an alternative strategy for alleviating colitis.

## 1. Introduction

Inflammatory bowel disease (IBD) is a group of disorders characterized by chronic inflammation of the intestinal tract [1], which has two main types: Crohn’s disease (CD) and ulcerative colitis (UC) [2,3]. The etiology of IBD is not well understood, and the pathogenesis and severity are influenced by genetic factors, immune response, intestinal microbiota, and oxidative stress [4]. In the development of IBD, excessive immune responses can lead to tissue damage and dysregulation of the intestinal microbiota, leading to metabolic disorders [5].

Drug therapy is commonly applied to alleviate IBD symptoms; however, no current strategy can completely cure IBD [6]. The clinical drugs for IBD mainly include salicylates [7], glucocorticoids [8], and immunosuppressive agents [9,10]. Salicylates (e.g., 5-aminosalicylic acid) reduce inflammation by downregulating intestinal proinflammatory factors. The decrease in proinflammatory factors decreases the induced inflammatory response. Salicylates are generally used in the treatment of patients with less severe IBD, but they may cause serious gastrointestinal and other adverse effects and have also been reported to cause kidney damage in patients [11]. Glucocorticoids (e.g., hydrocortisone and prednisone) inhibit the release of inflammatory substances (e.g., prostaglandins and leukotrienes) [12]. Patients with severe enteritis symptoms are usually treated with glucocorticoids. Glucocorticoids are likely to cause reduced effectiveness and some side effects when taken in large amounts [13]. Immunosuppressive agents (such as anti-tumor necrosis factor-α, anti-TNF-α [14]) can suppress the proliferation of inflammatory cells [15], and they are generally used in the treatment of patients with Crohn’s disease (CD) [16]. 

The incidence of UC is increasing worldwide, so there is an urgent need to develop safe and effective measures for UC prevention and treatment. The use of lactic acid bacteria (LAB) to ferment fruit and vegetable juices is currently a hot research topic. Lactic acid fermentation is a common method of fermentation. Organic acids, special enzymes, and other substances produced by lactic acid bacteria have special physiological functions. Many studies have also demonstrated that some lactic acid bacteria with probiotic functions and fermented products have antioxidant and anti-inflammatory properties and have been shown to improve UC [17]. For example, fermented egg-milk beverages could modulate intestinal microbiota and short-chain fatty acids, thus reducing colitis in mice [18]. Fermented *Astragalus* and its metabolites can regulate intestinal microbiota to repair intestinal barrier damage, thus reducing ulcerative colitis [19].

*Sargassum fusiforme* (*S. fusiforme*), also known as sea barley, sea sprouts, etc., is a sea vegetable belonging to the family of *Sargassaceae* in the class of *Phaeophyceae* [20]. It has been reported that *S. fusiforme* extract (the active ingredient) displays hypoglycemic, antitumor, hypolipidemic, and antioxidant effects [21]. For example, extracts of *S. fusiforme* had a significant hypoglycemic effect on type 2 diabetic rats and improved hyperlipidemia in type 2 diabetic rats [22]. The active ingredients of *S. fusiforme* can also suppress tumors by activating the regulatory mechanisms of tumor cells [23]. The crude extract of *S. fusiforme* has antioxidant activity by scavenging free radicals in the free radical diphenylpicrylhydrazyl (DPPH) system [24].

Nevertheless, there have been few reports on fermented *S. fusiforme*’s function in ulcerative colitis. Therefore, the aim of this study was to examine the effect of fermented *S. fusiforme* on dextran sulfate sodium (DSS)-induced ulcerative colitis in mice and potential mechanisms.

## 2. Experimental Methods

### 2.1. Preparation of Fermented S. fusiforme

One kilogram of dried *S. fusiforme* was purchased from Zhejiang Jinhaiyun Biotechnology Co., Ltd. (Wenzhou, China). The original *S. fusiforme* was harvested from the East Sea in April 2021 and then vacuum oven dried. The dried *S. fusiforme* was ground into powder with a blender in the laboratory and sieved to achieve an 80/100 mesh size. The powder was mixed with sterile water to form a *S. fusiforme* raw stock with a mass ratio of 1:10. Subsequently, the *S. fusiforme* stock solution was sterilized and cooled to room temperature after sterilization. *Lactobacillus acidophilus* ATCC 6005 was inoculated at a 1% inoculation rate in MRS liquid medium at 37 °C for 1 day and then activated for 3 generations for fermentation. Then the bacteria were inoculated into the stock solution of *S. fusiforme* at a 10% inoculation rate under sterile conditions, and the fermenter was sealed and placed in a 37 °C incubator for static fermentation for 2 days. Then, the broth was filtered to obtain fermented *S. fusiforme*. After being pasteurized, a part of the fermented *S. fusiforme* was frozen at −20 °C, and the other part was stored at 4 °C for further physicochemical testing and animal experiments.

### 2.2. Determination of the Physicochemical Properties of Fermented S. fusiforme

The total and reducing sugars of fermented *S. fusiforme* were determined by the dinitrosalicylic acid method. Lowry’s method was used to measure the phenolic compounds in fermented *S. fusiforme*. The α-amylase activity was measured by an alpha amylase kit. Superoxide dismutase (SOD) activity and total antioxidant capacity (T-AOC) were measured by SOD and T-AOC kits. Fermented *S. fusiforme* was treated according to the method of Xu et al. [25] and analyzed using a Hitachi LA8080 Amino Acid Analyzer (LA8080, Tokyo, Japan) fitted with a Hitachi high-performance cation-exchange column with a 57 °C column temperature. The flow rates of pump 1 and pump 2 were 0.400 mL/min and 0.350 mL/min, respectively. The detection wavelength was 570 nm, and the injection volume was 20 μL. All samples were deproteinized and centrifuged at 10,000× *g* for 10 min (CR22N, Hitachi, Tokyo, Japan) to remove the insoluble substances. An equal volume of acetone was added, followed by centrifugation at 10,000× *g* for 10 min to remove large proteins. After adding 0.02 M HCl, the samples were filtered using a 0.45 μm syringe filter.

### 2.3. Animal Experiments

#### 2.3.1. Experimental Animals

Fifty specific pathogen-free (SPF) male C57BL/6J mice (5 weeks old) were purchased from Liaoning Changsheng Biotechnology Co., Ltd. (Benxi, China). Under standard laboratory conditions, 7~8 mice in each cage were fed maintenance feed for mice (Liaoning Changsheng Biotechnology Co., Ltd.). All experiments were conducted following the guidelines for Experimental Animals of the National Institutes of Health. It was approved by the Animal Ethics Committee of Dalian Polytechnic University (Approval No. DLPU2021049).

#### 2.3.2. Mice Colitis Model and Dietary Intervention

Dextran sulfate sodium (DSS, purity of 98% and molecular weight of 36–50 kDa) was used to construct a mouse model of ulcerative colitis. After adaptive feeding, mice were randomly grouped into the control group, the DSS group, the F + DSS (fermented *S. fusiforme*) group, the W + DSS (unfermented *S. fusiforme*) group, and the Y + DSS (positive control) group. During the week of adaptive feeding, all five groups of mice drank and ate freely. For the first two weeks of the experiment, mice in the F + DSS group were orally gavaged with 200 μL of fermented *S. fusiforme*, mice in the W + DSS group were orally gavaged with 200 μL of unfermented *S. fusiforme,* and mice in the control, DSS, and Y + DSS groups were gavaged with equal amounts of PBS (Figure 1). Starting from day 1, mice in the DSS group, the F + DSS (fermented *S. fusiforme*) group, the W + DSS (unfermented *S. fusiforme*) group, and the Y + DSS (positive control) group started to drink 2.5% DSS water for seven days (Figure 1). During these seven days, we used mice gavaged with PBS as the control group and mice gavaged with 5-amino salicylic acid (5-ASA) as the positive control group. Five mice were raised in each rectangular cage under standard laboratory conditions (temperature 23 ± 2 °C, relative humidity 55 ± 5%, and 12 h light/dark cycle) and fed an SPF rodent maintenance feed (Liaoning Changsheng Biotechnology Co., Ltd.; no.GOA6FVYA1S10R4123) [25]. The mice were monitored for symptoms during the seven days of dietary intervention. The mice were euthanized after ten days. Plasma, colonic tissues, and intestinal contents were collected. Some colonic tissues were placed in 4% phosphate buffer (*v*/*v*) for fixation to facilitate further analysis.

### 2.4. Disease Activity Index (DAI) in Mice

DAI was assessed and recorded for each mouse daily during these seven days. The rules for scoring the Disease Activity Index (DAI) are shown in Table 1.

### 2.5. Histological Analysis

The 4% paraformaldehyde-fixed colonic tissues were paraffin-embedded. The treated colon was then cut into 3–5 µm slices and stained. PAS staining (periodic acid-Schiff staining) was used to observe the thickness of the intestinal mucosal epithelium and the goblet cells. To assess the intestinal mucosal epithelial thickness and the presence of goblet cells, periodic acid-Schiff staining (PAS) on the tissue sections was performed and analyzed under a light microscope, which was divided into 4 levels, with higher pathological scores indicating more severe mucosal inflammation. Zero: no obvious pathological changes; 1 point: low-level lymphocytic infiltration with no destruction of intestinal villous structures in a 10% high magnification field; 2 points: 10–25% high magnification, moderate lymphocytic infiltration, elongation of intestinal crypts, thickening of the intestinal wall without invasion of the muscular layer, no ulcer; 3 points: 25–50% high magnification, high level of lymphocytic infiltration, vascular hyperplasia, thickening of the intestinal wall, and invasion of the muscular layer, no ulcer; and 4 points: significant lymphocytic infiltration, vascular hyperplasia with an elongation of intestinal crypts, thickening of the intestinal wall, and invasion of the muscular layer, ulcer. Ulceration is seen [26].

### 2.6. In Vivo Intestinal Permeability Measurement

The mice were first fasted overnight and then orally gavaged with 200 μL of fluorescein isothiocyanate-dextran (FITC-D, 0.6 mg/g). Blood was taken from mice five hours after gavage. The plasma was centrifuged under light-proof conditions. The centrifugation conditions were 3000× *g* for 15 min. The serum was then obtained. A fluorescent enzyme marker was used to assess the amount of FITC-D in the blood [27].

### 2.7. Inflammatory Cytokine (IL-6, IL-8, TNF-α, and IL-1β) Assay and Western Blot Analysis

The collected blood of mice was centrifuged at 4 °C, and the supernatant was utilized for determination. The levels of proinflammatory cytokines, including IL-6, IL-8, TNF-α, and IL-1β, in the serum and colon were determined by using enzyme-linked immunosorbent assay (ELISA) kits (Shanghai Enzyme-linked Biotechnology Co., Ltd., Shanghai, China).

The prepared colon tissues were treated with radioimmunoprecipitation assay (RIPA) buffer containing protease inhibitors at 4 °C to extract proteins from the colon tissues. The concentration of colonic extract protein was determined, and the dilution to the appropriate concentration was carried out. Immunoblotting was carried out with slight modifications, according to Cao et al. [28]. A concentration of 8% and 10% polyacrylamide gels was selected according to the size of the separated target proteins, and the proteins were transferred to nitrocellulose membranes after transmigration through an electrophoresis apparatus. The proteins were transferred to a nitrocellulose membrane with skim milk powder, and the membrane was then incubated at 4 °C with primary antibodies against IκB-α, p65, p-p65, p-IκBα, zonulin-1 (ZO-1), occluding, and β-actin. The membranes were incubated with primary antibody diluent and then removed and washed with washing solution. The membranes were then placed in the secondary antibody diluent to completely submerge them and incubated for at least 2 h. Finally, the membranes were analyzed by a chemical imaging analysis system (ChemiDoc Touch, BIO-RAD, Hercules, CA, USA), and protein quantification was performed using the NIH ImageJ program. Relative expression of occludin and ZO-1 in the colon tissue was assessed by Western blot using β-actin as a control.

### 2.8. Nitric Oxide (NO) and Myeloperoxidase (MPO) Levels in Mice Serum

The colon tissue was weighed and mixed in with 0.1 M PBS (pH 7.4). The reagents were added following the kit’s instructions. Once the reaction was complete, the precipitate was centrifuged and discarded. A BCA kit was used to determine the protein concentration of the sample. As a final step, the samples were mixed with the kit reagents and boiled for 30 min at 37 °C. Finally, the 560 nm absorbance value was measured using an ultraviolet spectrophotometer (UV-5100B, Shanghai Yuan Analysis Instrument Analysis Company, Shanghai, China) [29].

### 2.9. Total Superoxide Dismutase (SOD) Enzyme Activity in Colonic Tissue

Colonic tissues were weighed and mixed with the solution from the kit (1:10, g/mL). The tissues and the solution were mixed thoroughly and then centrifuged (8000× *g*, 4 °C). The supernatant was collected after ten minutes of centrifugation. The manufacturer’s instructions were followed when determining the SOD enzyme activity.

### 2.10. The Colon Tissue for Malondialdehyde (MDA)

Colonic tissues were weighed and mixed with the solution from the kit (1:10, g/mL). The tissues and the solution were mixed thoroughly and then centrifuged (8000× *g*, 4 °C). The supernatant was collected after ten minutes of centrifugation. The concentration of colonic protein was determined. Then different reagents were added according to the kit instructions. The samples were placed in a boiling water bath for 1 h before absorbance measurements were performed.

### 2.11. Measurement of Catalase (CAT) in Serum and Colonic Tissues

Colonic tissues were weighed and mixed with the solution from the kit (1:10, g/mL). The tissues and the solution were mixed thoroughly and then centrifuged (8000× *g*, 4 °C). The supernatant was collected after ten minutes of centrifugation. The sample and working solution were immediately mixed and reacted at a ratio of 1:19. A 240 nm absorbance measurement was performed. The absorbance value was measured again after one minute for further calculation. The mouse serum was assayed directly according to the above procedure, and the initial absorbance value at 240 nm and the absorbance value after 1 min were recorded.

### 2.12. Detection of SCFA Content in Feces

The cecum contents frozen in advance were removed, and 0.1 g of each mouse was weighed out and set aside. The cecum contents were acidified with dilute sulfuric acid, and then ether and 2-ethyl butyric acid were added in a ratio of 9:1. The mixture was then treated as in the study of Cao et al. [30]. The mixture was sonicated at 4 °C for one hour. The mixture was then centrifuged at 10,000× *g* at 4 °C for twenty minutes, and the supernatant was retained. Finally, SCFAs (acetate, propionate, butyrate, isobutyrate, valerate, and isovalerate) in feces were analyzed by gas chromatography (Shimadzu GC2010-plus, Kyoto, Japan).

### 2.13. Statistical Analysis

Data are expressed as the mean ± standard deviation. Statistical analysis was performed using GraphPad Prism 9.00 software. Statistical significance was determined by one-way analysis of variance (ANOVA). The difference was considered statistically significant when *p* < 0.05.

## 3. Results

### 3.1. Physicochemical Properties of Fermented S. fusiforme

Fermented *S. fusiforme* was prepared following the procedure described in the Methods section. The protein content in fermented *S. fusiforme* was 0.272 mg/mL, the phenol content was 0.058 mg/mL, the acid content was 2.18‰, the reducing sugar content was 0.301%, the total sugar content was 7.57%, and the α-amylase activity was 40.3 U/dL. In terms of antioxidant properties, fermented *S. fusiforme has an* SOD enzyme activity of 19.3 U/mL and a total antioxidant capacity of 0.37 μmol/mL. In addition, free amino acids in fermented *S. fusiforme* were detected by an amino acid analyzer, and the results showed that some new amino acids appeared after fermentation, including aspartic acid, threonine, serine, glycine, valine, isoleucine, leucine, tyrosine, lysine, and proline (Figure 2). After fermentation, the concentration of alanine increased from 0.054 nmol to 0.824 nmol, and the concentration of phenylalanine increased from 0.133 nmol to 0.290 nmol.

### 3.2. Fermented S. fusiforme Alleviated Symptoms of Colitis in Mice

The control group of mice had a stable body weight. The mice in all other groups lost weight. However, the fermented *S. fusiforme* group, the unfermented *S. fusiforme* group, and the positive control group alleviated the trend of weight loss to different degrees. Compared to mice treated only with DSS, mice treated with fermented and unfermented *S. fusiforme* lost less weight after DSS treatment (Figure 3A). On day 7, low-field magnetic resonance detection was performed on each group of mice [31]. The results showed that the water content in the intestines of the mice in the fermented *S. fusiforme* group was less than that in the DSS group (Figure 3B). Fermented *S. fusiforme* reduces weight loss due to DSS. The control mice had the lowest DAI scores, and the DAI scores of mice in the positive control, fermented *S. fusiforme*, and unfermented *S. fusiforme* groups gradually increased (Figure 3C). Compared to mice in the colitis model group, mice in the three intervention groups lost less weight and showed less frequent symptoms such as rectal bleeding. The colon length of mice in the fermented *S. fusiforme*, unfermented *S. fusiforme*, and positive control groups was longer than that in the DSS group and shorter than that in the control group. This indicates that the interventions all had an effect on reducing colonic shortening, but to a different extent (Figure 3D). The mice in the control group had normal colonic tissue color without congestion. The colonic tissue was of normal thickness without edema and adhered to the surrounding tissues (Figure 3E). The DSS group mice showed congestion and swelling of the colon, and their colon was significantly shortened. The shortening of the colon was reduced in mice in the fermented *S. fusiforme* group, the unfermented *S. fusiforme* group, and the positive control group.

### 3.3. Fermented S. fusiforme Attenuated Histological Damage Caused by DSS

In mice in the control group, the epithelial cells were structurally intact, without necrosis, and a high number of goblet cells could be seen, with abundant and well-arranged intestinal glands. The gaps in the submucosal layer were uniform in size and had no pathological features. Mice in the DSS group had necrosis of the mucosal layer, loss of crypt structures, and diffuse infiltration of inflammatory cells. The connective tissue in the submucosal layer proliferated, and the tissue gap was enlarged; the muscle fiber gap was locally enlarged, the fibers were loosely arranged, and the connective tissue between fibers proliferated. The histological lesions were improved in the fermented *S. fusiforme* intervention group and the positive drug intervention group (Figure 4A). According to Figure 4C, the pathological scores of mice in the DSS model group were significantly higher than those in the control group, while the intervention by the F + DSS, W + DSS, and Y + DSS groups was able to significantly reduce the pathological scores of mice (*p* < 0.05), with the highest reduction in the F + DSS group, which had a similar effect to Y + DSS. These data indicate that fermented *S. fusiforme* can effectively alleviate DSS-induced colitis.

### 3.4. Inhibition of Permeability Increase of Colonic Epithelium in Colitis by Fermented S. fusiforme

Fluorescein isothiocyanate-dextran (FITC-D) was used to evaluate the integrity of the colonic epithelium. The FITC-D concentration in plasma was determined 4 h after administration. Plasma levels of FITC-D were higher in mice in the DSS group, whereas the fermented *S. fusiforme* intervention significantly decreased them (Figure 4B). This indicates that fermented *S. fusiforme* has an inhibitory effect on the DSS-induced increase in colonic epithelial permeability in mice with colitis.

### 3.5. Protective Effect of Fermented S. fusiforme on DSS-Induced Tight Junction Proteins in Mice with Colitis

Zonula occluden-1 (ZO-1) and occludin, as markers of epithelial integrity, are important epithelial tight junction proteins whose role is to maintain the intestinal mechanical barrier. DSS was able to significantly reduce the transcript levels of ZO-1 and occludin. Compared with the DSS group, all intervention groups exhibited increased transcript levels of tight junction proteins to different degrees (Figure 5A). The F + DSS and Y + DSS groups significantly increased ZO-1 and occludin transcript levels, while the W + DSS group failed to increase ZO-1 and occludin transcript levels (Figure 5B,C). These data indicate that fermented *S. fusiforme* could increase colonic tight junction protein levels and improve mouse intestinal barrier function.

### 3.6. Fermented Sargassum Fusiforme Decreased Levels of Inflammatory Cytokines and the Inhibited NF-κB Pathway

Cytokines serve as molecular messengers with regulatory and effector functions in many diseases. Thus, cytokines can stimulate the production of proinflammatory mediators and activate inflammatory pathways, thereby affecting the inflammatory response in patients with IBD.

To investigate how fermented *S. fusiforme* affected ulcerative colitis, inflammatory cytokines were measured. An important cause of damage to the colon is the massive release of proinflammatory cytokines. The determination of IL-6, IL-8, TNF-α, and IL-1β in mice serum revealed that the levels of proinflammatory factors in the DSS model mice were significantly higher than those in the control normal mice. The serum levels of proinflammatory factors in the F + DSS and Y + DSS groups were significantly lower than those in the DSS group. However, only the proinflammatory factors IL-8 and TNF-α were significantly decreased in the serum of the W + DSS group, while IL-6 and IL-1β did not change significantly (Figure 6A–D). This indicates that the fermented *S. fusiforme* intervention group can inhibit the secretion of the proinflammatory factors IL-6, IL-8, TNF-α, and IL-1β in mice serum.

Many studies have confirmed the involvement of the NF-κB signaling pathway in a variety of inflammatory diseases. NF-κB is an important class of transcription factors that is involved in the transcriptional control of several cytokines related to immunological inflammation and is crucial for the development of inflammation. Extracellular stimulation induces the NF-κB signaling pathway, causing its receptor protein to activate IκB kinase (IKK) in response to stimulation. IKK phosphorylates amino acids, which release NF-κB dimers. NF-κB dimers enter the nucleus to bind to IBD-related genes and initiate the transcriptional process that leads to inflammation. The positive drug intervention group (Y + DSS), fermented *S. fusiforme* (F + DSS), and control groups showed significantly less phosphorylation of p65 and IκB-α. Unfermented *S. fusiforme* only decreased the phosphorylated protein expression of p65, while the phosphorylated protein expression of IκB-α increased (Figure 7A–C).

### 3.7. Fermented S. fusiforme Reduced Oxidative Stress

Oxidative stress and the inflammatory response can interact with each other. To study the effect of fermented *S. fusiforme* on oxidative stress, nitric oxide levels (NO), myeloperoxidase activity (MPO), total superoxide dismutase activity (T-SOD), catalase activity (CAT), and malondialdehyde (MDA) levels were measured in colonic tissues. In addition, peroxidase activity (CAT) was also measured in mouse serum. In general, an increase in NO concentration and MPO activity in colonic tissue leads to an increased degree of colonic inflammation. Among all groups of mice, fermented *S. fusiforme*, unfermented *S. fusiforme*, and the positive control group significantly reduced the NO level in colitis mice. However, only the fermented *S. fusiforme* and positive control groups showed significant reductions in MPO levels, while no significant differences were observed in mice in the W + DSS group (Figure 8A,B). SOD plays an important role in antioxidant action in biological organisms. The DSS model group significantly inhibited the activity of T-SOD (Figure 8C) in the colon, and the CAT levels in both serum and colonic tissues of mice were decreased in the DSS model group (Figure 8D,E). The MDA concentration in the colon of mice in the DSS model group was increased (Figure 8F). The F + DSS and Y + DSS groups showed an increase in T-SOD activity and CAT levels and a decrease in MDA concentrations (Figure 8C–F) compared to the DSS group.

### 3.8. Restoration of SCFA Production in the Cecum by Fermented S. fusiforme

IBD is associated with changes in SCFAs in the intestine. Measurements were made on acetate, propionate, butyrate, isobutyrate, valerate, and isovalerate in mice. The mice in the DSS model group contained significantly lower levels of SCFA than normal control mice. Only the level of propionate was considerably higher in the unfermented *S. fusiforme* group than in the DSS model group; the content of other SCFAs was almost identical to that of the DSS model group. However, in both the fermented *S. fusiforme* (F + DSS) and positive control (Y + DSS) groups, there was a trend toward a significant increase in SCFA production, especially propionate, butyrate, valerate, and isovalerate (Figure 9A–F). Fermented *S. fusiforme* increased SCFAs in colitic mice as measured by the short-chain fatty acid assay.

## 4. Discussion

Currently, anti-inflammatory or immunosuppressive drugs such as aminosalicylic acid, corticosteroids, and thiopurines are mainly used to treat IBD [32]. However, these drugs have certain serious side effects on human health. In recent years, there have been alternative or complementary strategies for IBD, such as probiotic supplementation (e.g., lactobacilli and bifidobacteria) [33] and herbal interventions [34] for patients with IBD. Many studies have found that fermented fruit and vegetable products have greater potential for the treatment of UC [19]. We examined the effects of *S. fusiforme* aqueous extract and fermented *S. fusiforme* on DSS-induced colitis, and the findings indicate that *S. fusiforme* fermented by *L. acidophilus* ATCC 6005 was more potent than *S. fusiforme* aqueous extract in improving colitis. Fermented *S. fusiforme* could mitigate ulcerative colitis in mice by regulating the intestinal barrier, oxidative stress, and the NF-κB pathway.

There is a physical barrier in the intestine that blocks harmful and toxic substances from entering the rest of the body from the intestine. This physical barrier is formed by the mucus layer and epithelial cells [35]. This barrier serves to protect the health of the intestine. HE staining of mouse colonic tissues revealed disruption of colonic epithelial integrity in the DSS model group of mice. Mice in the fermented *S. fusiforme* intervention and positive drug intervention groups had reduced colonic epithelial cell injury. A reduction in epithelial barrier function can also be attributed to the protein molecules ZO-1, occludin, and claudin-1 [36]. The ZO-1 protein plays an important role in regulating epithelial integrity, especially in tight junctions [37]. Occludin has an important role in tight junction stability and barrier function, which are essential for maintaining intestinal integrity [38]. ZO-1 and occludin expression were reduced in mice in the DSS model group, indicating that intestinal integrity was disrupted. The aqueous extract of *S. fusiforme* failed to enhance the expression of the protein molecules ZO-1 and occludin. However, intervention with fermented *S. fusiforme* effectively increased the expression of the protein molecules ZO-1 and occludin, thereby protecting the integrity of the mouse intestine. Fermented *S. fusiforme* was also able to reduce FITC-D levels in mouse serum, which indicates a decrease in intestinal permeability.

IBD is linked to the NF-κB signaling pathway [39]. Therefore, the expression of the proteins p65 and IκB-α and their phosphorylation were determined in this study. It was found that unfermented *S. fusiforme* failed to improve colitis through the NF-κB signaling pathway, while fermented *S. fusiforme* did. This study examined the levels of inflammatory factors in the serum of mice. TNF-α is elevated at both local and systemic levels in the intestine of patients with IBD [40]. Excessive TNF-α could increase mucosal inflammation, damage the intestinal barrier, reduce tight junction function, and increase permeability as well as apoptosis of intestinal epithelial cells [41]. Excessive TNF-α could also induce the secretion of other cytokines and adhesion molecules. Abnormal secretion of IL-1β induces the release of other inflammatory factors and aggravates the local inflammatory response in the colonic mucosa. IL-6 activates intestinal target cells, promotes the inflammatory response, prevents apoptosis of mucosal T cells, and activates these cells to produce inflammatory factors [42]. DSS treatment resulted in increased amounts of the proinflammatory factors IL-6, TNF-α, IL-1β, and IL-8 in the colon of mice. The aqueous extract of *S. fusiforme* showed only a small reduction in the inflammatory factors TNF-α and IL-8. However, both the fermented *S. fusiforme* intervention and the positive drug intervention reduced the amount of proinflammatory factors in the colons of mice.

Oxidative stress is a component of the inflammatory response [43], which leads to an imbalance between oxidative and antioxidant effects in the body and the production of large amounts of oxidative products. Many of the substances produced by oxidative stress are proinflammatory and thus induce an inflammatory response [43]. Reactive oxygen species are produced in significant quantities as a result of inflammatory diseases, which encourage the release of proinflammatory cytokines from cells [44]. Excessive ROS can lead to oxidative stress in tissues. The formation of oxygen radicals increases when NO levels rise, which is linked to the onset of ulcerative colitis. Oxygen-free radicals cause oxidative stress in tissues, which in turn induces inflammation [45]. DSS treatment resulted in significantly higher NO levels in mouse colonic tissue. The colonic NO concentration was significantly reduced in the unfermented *S. fusiforme* group, and the reduction was even greater in the fermented *S. fusiforme* group. In addition, MPO catalyzes oxidative reactions, resulting in oxidative stress. Oxidative stress, in turn, interacts with the inflammatory response. Therefore, MPO and ulcerative colitis are closely related. DSS stimulation resulted in a significant increase in MPO activity in mouse colonic tissue. Unfermented *S. fusiforme* failed to reduce MPO activity in colonic tissue, which was almost the same as in the DSS group. However, the fermented *S. fusiforme* intervention group significantly reduced MPO activity in colonic tissue. In addition, fermented *S. fusiforme* was able to increase the amount of the antioxidant enzymes CAT and T-SOD and reduce the production of MDA.

In this study, although there was an indication of improvement in colitis and certain parameters in mice treated with *S. fusiforme* aqueous extract, fermented *S. fusiforme* had a more effective outcome on UC. This may be due to the ability of *L. acidophilus* to produce abundant enzyme systems utilizing substances such as polysaccharides in *S. fusiforme*. Meanwhile, the fermentation of *S. fusiforme* converts large molecules into small metabolites. The human body can more easily absorb small metabolites, thus further enhancing the therapeutic effects of *S. fusiforme*. In addition, the metabolites produced by lactic acid bacteria may have synergistic effects with the active substances in *S. fusiforme*. Therefore, the metabolites produced by fermented *S. fusiforme* necessitate further investigation.

According to several studies, patients with IBD had lower levels of the bacteria that produce SCFAs (*Faecalibacterium prausnitzii* and *Roseburia intestinalis*) in their mucosa and feces compared to healthy people [46,47]. This finding raises the possibility that SCFAs and IBD are strongly related. In the present study, unfermented *S. fusiforme* failed to increase the number of SCFAs in the cecum, while fermented *S. fusiforme* increased the number of SCFAs in the cecum. We hypothesize that fermented *S. fusiforme* may reduce colitis by improving the intestinal microbiota environment, and follow-up trials are needed to confirm this hypothesis.

## 5. Conclusions

Fermented *S. fusiforme* could effectively alleviate DSS-induced colitis, which is associated with enhanced tight junctions, reduced expression of proinflammatory factors, decreased oxidative stress, and restored levels of SCFAs. These findings suggest that fermented *S. fusiforme* might be utilized as a functional food or complementary strategy for alleviating IBD. However, human interventional studies are necessary to elucidate the benefits of fermented *S. fusiforme*.

## Figures and Tables

**Figure 1 foods-12-01928-f001:**
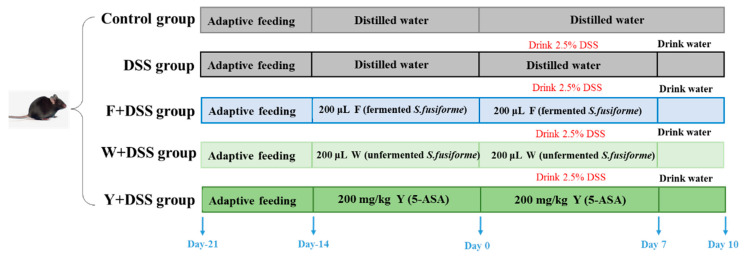
Schematic depicting the experimental design. Mice were divided into five groups and treated differently as indicated.

**Figure 2 foods-12-01928-f002:**
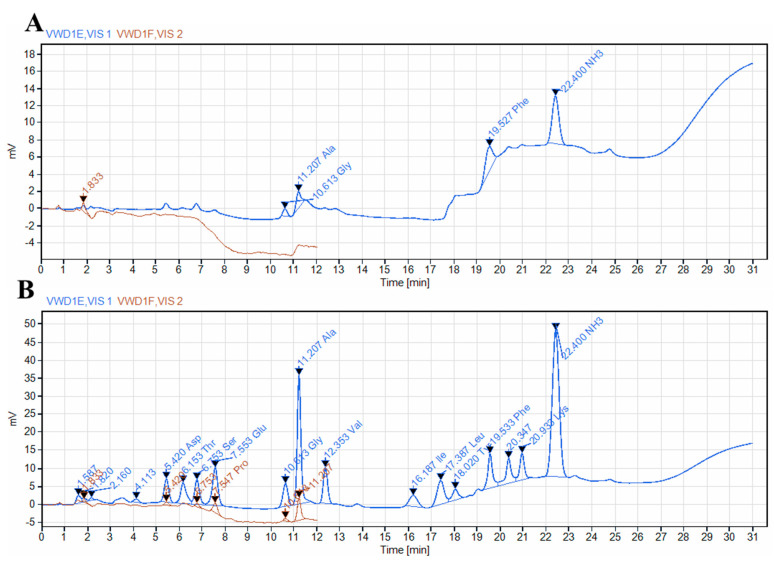
The amino acid profile of (**A**) unfermented *S. fusiforme* and (**B**) fermented *S. fusiforme*.

**Figure 3 foods-12-01928-f003:**
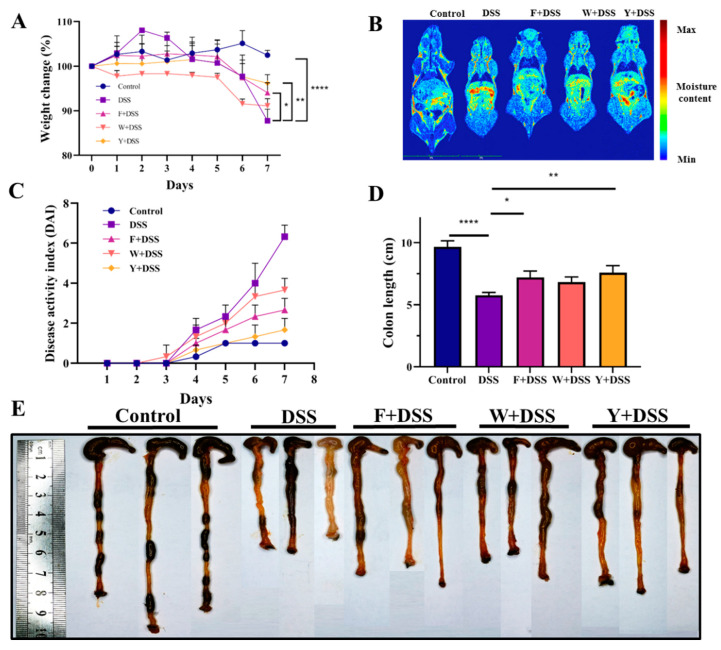
Fermented *S. fusiforme* alleviated the clinical symptoms of colitis. (**A**) Body weight change (%). (**B**) Low-field magnetic resonance images illustrating the body morphology and water accumulation of mice. (**C**) Disease activity index (DAI). (**D**) Colon length of mice (cm). (**E**) Representative images of the colon. The F + DSS group was the fermented *S. fusiforme* intervention group, the W + DSS group was the unfermented *S. fusiforme* intervention group, and the Y + DSS group was the positive drug intervention group. * *p* < 0.05, ** *p* < 0.01, **** *p* < 0.0001.

**Figure 4 foods-12-01928-f004:**
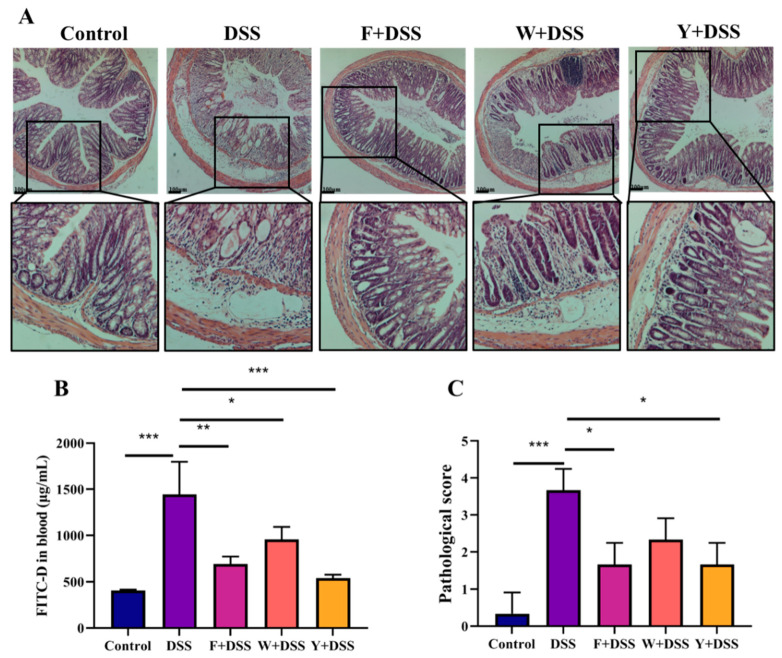
Fermented *S. fusiforme* mitigated colon tissue damage in colitis mice. (**A**) Hematoxylin and eosin-stained sections showing the condition of the colon in different groups of mice. (**B**) An evaluation of intestinal permeability was carried out by detecting FITC-D in the blood. (**C**) Pathological scores of mice in different groups. The F + DSS group was the fermented *S. fusiforme* intervention group, the W + DSS group was the unfermented *S. fusiforme* intervention group, and the Y + DSS group was the positive drug intervention group. * *p* < 0.05, ** *p* < 0.01, *** *p* < 0.001.

**Figure 5 foods-12-01928-f005:**
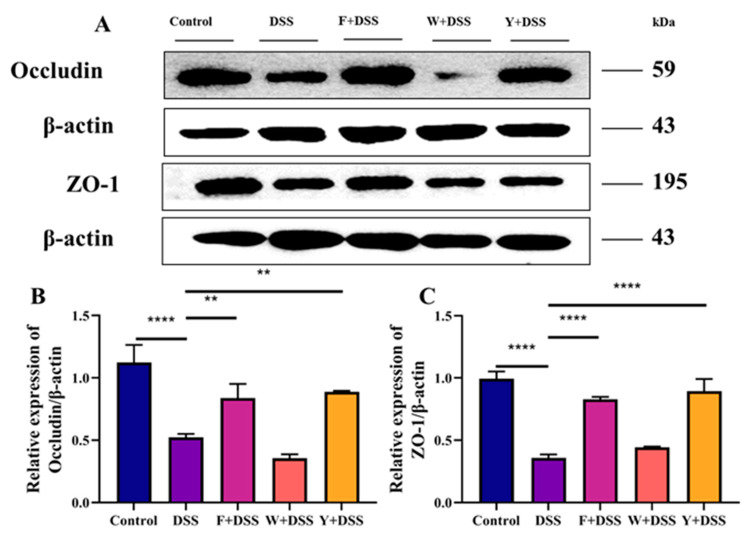
Fermented *S. fusiforme* improved colonic epithelial integrity, tight junction proteins, and mucin expression. (**A**) Western blotting was used to determine the relative expression of occludin and ZO-1 in colon tissue, using β-actin as a control. Using the NIH ImageJ program, a bar graph displaying the relative band intensities of the occludin (**B**) and ZO-1 (**C**) proteins versus β-actin as the internal control was generated. The F + DSS group was the fermented *S. fusiforme* intervention group, the W + DSS group was the unfermented *S. fusiforme* intervention group, and the Y + DSS group was the positive drug intervention group. ** *p* < 0.01 and *****p* < 0.0001.

**Figure 6 foods-12-01928-f006:**
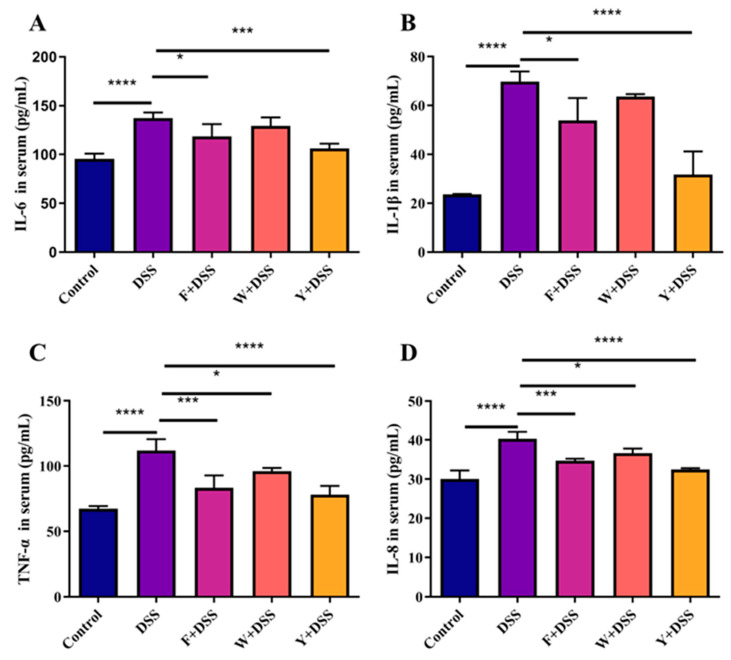
Effect of fermented *S. fusiforme* on proinflammatory cytokines in colitic mice. (**A**) IL-6, (**B**) IL-1β, (**C**) TNF-α, and (**D**) IL-8 levels in the serum. The F + DSS group was the fermented *S. fusiforme* intervention group, the W + DSS group was the unfermented *S. fusiforme* intervention group, and the Y + DSS group was the positive drug intervention group. * *p* < 0.05, *** *p*< 0.001, and *****p* < 0.0001.

**Figure 7 foods-12-01928-f007:**
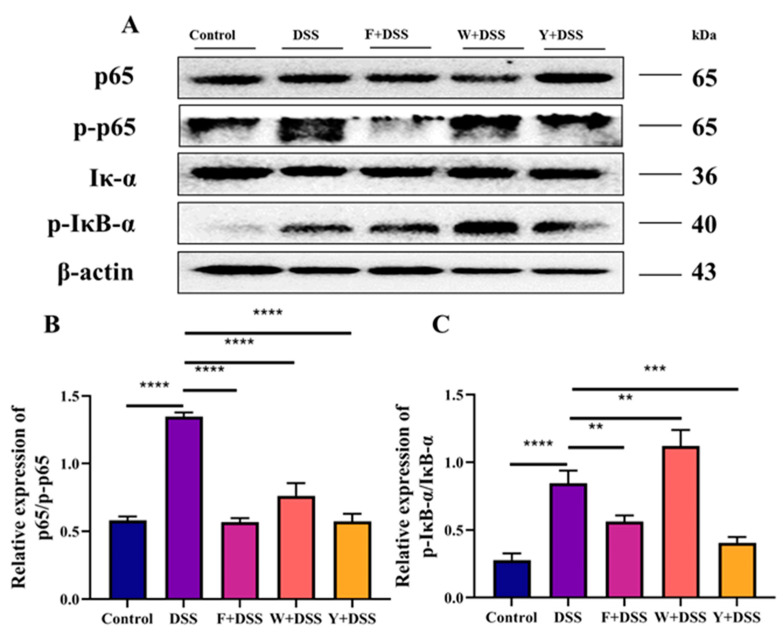
Effect of fermented *S. fusiforme* on the NF-κB signaling pathway. (**A**) Western blot images using β-actin as an internal control showed the relative expression of the p65, p-p65, Iκ-B-α, and p-IκB-α proteins. (**B**,**C**) Bar graphs displaying the relative band intensity of each protein as measured by the NIH ImageJ program. The F + DSS group was the fermented *S. fusiforme* intervention group, the W + DSS group was the unfermented *S. fusiforme* intervention group, and the Y + DSS group was the positive drug intervention group. ** *p* <0.01, *** *p* < 0.001, and *****p* < 0.0001.

**Figure 8 foods-12-01928-f008:**
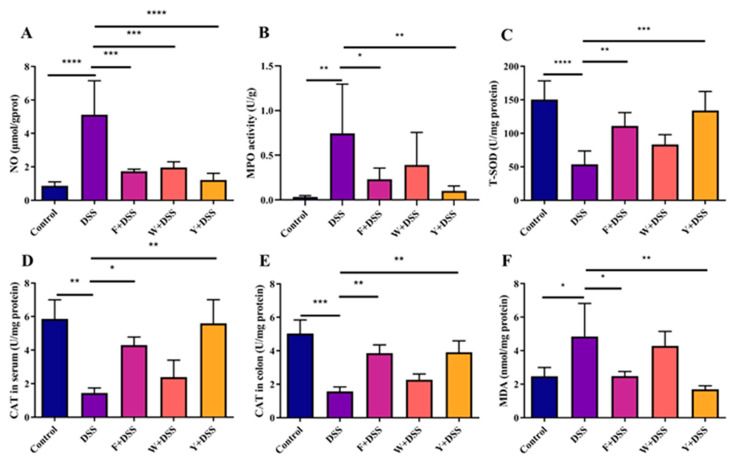
Fermented *S. fusiforme* ameliorated oxidative stress. Concentrations of NO activity (**A**), MPO (**B**), T-SOD (**C**), CAT (**E**), and MDA (**F**) in the colon (*n* = 6 and 7). Concentrations of CAT activity (**D**) in the serum (*n* = 6 and 7). The F + DSS group was the fermented *S. fusiforme* intervention group, the W + DSS group was the unfermented *S. fusiforme* intervention group, and the Y + DSS group was the positive drug intervention group. **p* < 0.05, ***p* < 0.01, ****p* < 0.001, and *****p* < 0.0001.

**Figure 9 foods-12-01928-f009:**
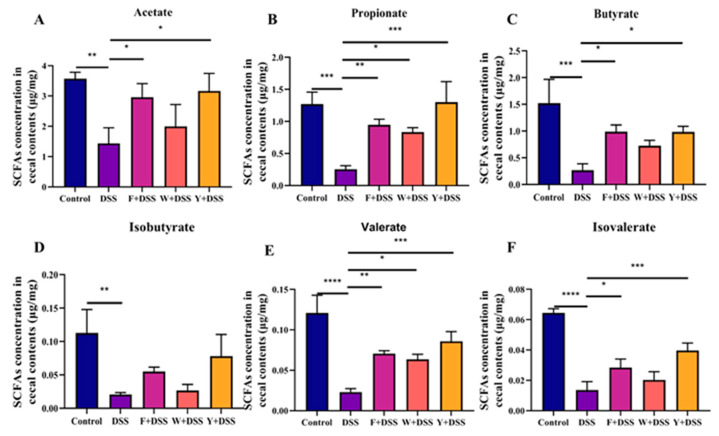
Fermented *S. fusiforme* ameliorated the decrease in SCFA levels in the cecum. The amounts of the following SCFAs were found in the cecal contents: (**A**) acetate, (**B**) propionate, (**C**) butyrate, (**D**) isobutyrate, (**E**) valerate, and (**F**) isovalerate. The F + DSS group was the fermented *S. fusiforme* intervention group, the W + DSS group was the unfermented *S. fusiforme* intervention group, and the Y + DSS group was the positive drug intervention group. * *p* < 0.05, ** *p* < 0.01, *** *p* <0.001, and *****p* < 0.0001.

**Table 1 foods-12-01928-t001:** Scoring system for the Disease Activity Index (DAI) of the colon.

Score	Weight Loss (%)	Stool Consistency	Occult/Gross Bleeding
0	None	Normal	Negative
1	1–5	Loose stool	Positive
2	5–10		
3	10–20		
4	>20	Diarrhea	Gross bleeding

## Data Availability

All related data and methods are presented in this paper. Additional inquiries should be addressed to the corresponding author.

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
