# Peer review of "Fermented Sargassum fusiforme Mitigates Ulcerative Colitis in Mice by Regulating the Intestinal Barrier, Oxidative Stress, and the NF-κB Pathway"

_foods, 2023, doi:10.3390/foods12101928_

Round 1
Reviewer 1 Report
The article contains interesting but controversial findings. It must meet the following critical corrections.
1. The introduction needs serious reworking. To describe the role of the lactic acid bacteria themselves and their relationship to the treatment of colitis, there are a huge number of articles on the subject. To edit the text to avoid multiple repetitions of Sargassum fusiforme in each sentence. After its first appearance, it may be abbreviated to S. fusiforme, or "it" or other substitution may be written.
2. Nowhere is it mentioned what the abbreviation DSS-induced ulcerative colitis means.
3. The types of lactic acid bacteria that were used in the fermentation, nor the fermentation conditions such as vessels, stirring, and aeration, are not specified at all. The species of lactic acid bacteria is of particular importance, as different species produce different liquid metabolites that can affect the whole study and therefore need to be discussed separately.
4. Also, it is not at all clear how large the pieces of algae are that are being fermented. This section needs to be rewritten.
5. In the discussion, it is necessary to distinguish which of the effects is due to the algae, and which - to the lactic acid bacteria, adding information about the influence of the bacteria themselves on the investigated processes. In fact, it is not clear whether the algae act more as a carrier and the main actor is the probiotic strains, or whether the composition of the algae itself leads to the healing effects. By bypassing the role of lactic acid bacteria the reader is left with the wrong impression reading the discussion.
Author Response
Dear Madam/Sir,
We appreciate the detailed and constructive comments provided by editor and the reviewers. We have carefully revised the manuscript by incorporating all the suggestions by the review panel. The details of our responses to the reviewers' comments are appended in the end of this letter. The manuscript are in the revised mode.
Thank you very much for all your help and look forward to hearing from you soon.
Best regards
Xiaodong Xia, PhD, Professor

Reviewer 2 Report
The manuscript describes an interesting study aimed at demonstrating the effect of fermented S. fusiforme in mitigating ulcerative colitis in mice. From the scientific point of view, the study is very well carried out, with adequate methodology, clear objectives and encouraging results. However, it shows several important omissions, it must to complete certain concepts and, in general, it should improve its formal presentation, not at the content level. Observations are detailed:
Title: "intestinal barrier", not in italics. Correct "intestianl"
- Always use italics to write genus and species of organisms. See References and p. 14, first paragraph
-line 65: write that it is "DSS"
- line 68: replace "fermentation" with "fermented"
- line 73: it is necessary to clarify which lactic acid bacteria were used. Was it a commercial starter?
-line 76: fermented at what temperature? specify
-2.3.2, several problems
. It should be clarified what "control" means (line 97)
. Fig 1- It is not understood what "Enzyme solution" means. Shouldn't it be spelled "fermented liquid"?
- 2.12: what is "RIPA"?
- 3.2: what is MRI?
- 3.4: what is FITC-D?
- Important: the results are well commented. The problem is that many comments that were already said in Results are repeated in Discussion, even all the Figures are mentioned. Enough to mention and analyze them in Results
- Conclusions, line 3rd, "...factors, decreased oxidative..."
Author Response

(The authors gave the same response as above.)

Round 2
Reviewer 1 Report
The text was improved. However, there are many typographical errors in the manuscript. For example, there should be a space before the square brackets everywhere, but not after them.
Author Response

(The authors gave the same response as above.)

Reviewer 2 Report
The authors have responded satisfactorily to the observations made. It only remains to review References since the genus and species of microorganisms must be written in italics
Author Response

(The authors gave the same response as above.)
